# Hydroxyl Radical vs. One-Electron Oxidation Reactivities in an Alternating GC Double-Stranded Oligonucleotide: A New Type Electron Hole Stabilization

**DOI:** 10.3390/biom13101493

**Published:** 2023-10-08

**Authors:** Annalisa Masi, Amedeo Capobianco, Krzysztof Bobrowski, Andrea Peluso, Chryssostomos Chatgilialoglu

**Affiliations:** 1Istituto per la Sintesi Organica e la Fotoreattività, Consiglio Nazionale delle Ricerche, 40129 Bologna, Italy; annalisa.masi@ic.cnr.it; 2Dipartimento di Chimica e Biologia “A. Zambelli”, Università di Salerno, 84084 Fisciano, Italy; acapobianco@unisa.it (A.C.); apeluso@unisa.it (A.P.); 3Centre of Radiation Research and Technology, Institute of Nuclear Chemistry and Technology, 03-195 Warsaw, Poland; kris@ichtj.pl; 4Center for Advanced Technologies, Adam Mickiewicz University, 61-614 Poznań, Poland

**Keywords:** DNA damage, oligonucleotide, hydroxyl radical, one-electron oxidation, electron hole stabilization, pulse radiolysis, DFT calculations, reaction mechanism

## Abstract

We examined the reaction of hydroxyl radicals (HO^•^) and sulfate radical anions (SO_4_^•−^), which is generated by ionizing radiation in aqueous solutions under anoxic conditions, with an alternating GC doubled-stranded oligodeoxynucleotide (ds-ODN), i.e., the palindromic 5′-d(GCGCGC)-3′. In particular, the optical spectra of the intermediate species and associated kinetic data in the range of ns to ms were obtained via pulse radiolysis. Computational studies by means of density functional theory (DFT) for structural and time-dependent DFT for spectroscopic features were performed on 5′-d(GCGC)-3′. Comprehensively, our results suggest the addition of HO^•^ to the G:C pair moiety, affording the [8-HO-G:C]^•^ detectable adduct. The previous reported spectra of one-electron oxidation of a variety of ds-ODN were assigned to [G(-H^+^):C]^•^ after deprotonation. Regarding 5′-d(GCGCGC)-3′ ds-ODN, the spectrum at 800 ns has a completely different spectral shape and kinetic behavior. By means of calculations, we assigned the species to [G:C/C:G]^•+^, in which the electron hole is predicted to be delocalized on the two stacked base pairs. This transient species was further hydrated to afford the [8-HO-G:C]^•^ detectable adduct. These remarkable findings suggest that the double-stranded alternating GC sequences allow for a new type of electron hole stabilization via delocalization over the whole sequence or part of it.

## 1. Introduction

Reactive oxygen species (ROS) generated as a result of normal intracellular metabolism include hydroxyl radicals (HO^•^) that can cause damage to biomolecules [1]. The reaction of HO^•^ with DNA is known to be deleterious in vivo, producing a variety of lesions that have been studied in detail from various points of view, including biophysical, biochemical, biological, and diagnostic approaches [2,3,4]. The nature of DNA lesions depends on the type of attack, i.e., the hydrogen atom abstraction from the 2′-deoxyribose units or the addition to the base moieties, with the latter being the predominant one (accounting for 85–90% of the attacked sites) [3,5]. The 8-oxo-7,8-dihydro-guanine moiety (8-oxo-G) is a major DNA lesion produced by oxidative stress, and its formation is the result of an initial HO^•^ attack or the one-electron oxidation step [3]. The mechanism of 8-oxo-G formation by the reaction of HO^•^ with calf thymus DNA [6,7] and 21-mer double-stranded oligonucleotides [8] in the presence of oxygen has been investigated in a few detailed studies. It has been proposed that three pathways contribute to the formation of the 8-oxo-G moiety, with the two major ones being oxygen-dependent [6,7,8]. The proposed mechanistic scheme reported in Figure 1 consists of directly adding HO^•^ to the C8 position of the guanine moiety, forming 8-HO-G^•^ as the minor path (5–10%) followed by one-electron oxidation. The majority of HO^•^ attacks (90–95%) will be unselective, producing a variety of carbon-centered radicals that, in the presence of O_2_, give the corresponding peroxyl radicals [5]. It has been suggested that 8-oxo-G is produced by two types of reactions in approximately equal amounts, with both involving DNA–OO^•^ radicals: (1) an intramolecular electron transfer leading to DNA–OO^−^ together with the Watson–Crick base pair radical cation [G:C]^•+^ and (2) intramolecular oxidation through the addition of peroxyl radicals onto the C8 of a vicinal guanine base [6,7]. Model studies on oligonucleotides have reported the transient generation of pyrimidine peroxyl radicals and their addition to the C8 of a vicinal guanine base [9,10]. It is also worth mentioning that Fapy-G lesions are chemically related to 8-oxo-G since they are generated from the same intermediate 8-HO-G^•^, and their formation depends on the oxygen concentration and the redox environment (Figure 1) [11,12,13].

The redox properties of nucleobases, nucleosides, nucleotides, and DNA sequences have been investigated via the use of several techniques [14,15,16,17]. Liquid jet photoelectron spectroscopy, voltammetry, and quantum chemical calculations find G as the most easily oxidizable nucleobase. The oxidation potential of adenine (A) is estimated to be 0.4 eV higher than guanine, while the pyrimidine moieties of cytosine and thymine (C,T) are oxidized at a potential higher than G (0.6–0.8 eV) [17]. Therefore, the G radical cation (G^•+^) is considered the putative initial intermediate in the oxidative DNA damage [17,18,19,20,21]. Furthermore, the formation of the Watson–Crick G:C and A:T pairs in chloroform solution lowers the oxidation potentials of G and A [22,23]; in oligonucleotides containing consecutive G and A homosequences, the formation of delocalized hole domains is possible and has been observed both via voltammetry and time-dependent spectroscopy [24,25]. Particular attention is given to the one-electron oxidation of the G:C pair and the complex mechanism of the deprotonation vs. hydration steps of [G:C]^•+^ pair (Figure 1) [26]. For the hydration reaction of [G:C]^•+^ in DNA, the identification of 8-HO-G^•^ by electron spin resonance (ESR) spectroscopy has been reported [27,28], and information on the reaction mechanism has also been reported [29]. To our knowledge, the water trapping rate of [G:C]^•+^ has not been measured yet, but a pseudo-first-order rate constant was estimated to be 6 × 10^4^ s^−1^ via an indirect method in [30]. Moreover, there have been numerous studies on the formation and behavior of G^•+^ in ds-ODNs using spectroscopic and product studies [31].

It is worth mentioning that the role of HO^•^ in redox processes with DNA is not well understood. Evidence has suggested that HO^•^ and one-electron oxidants may partly induce common degradation pathways [32]. Very often, the description of this chemistry is simplified by reporting the known chemistry of corresponding nucleosides. However, the reaction of HO^•^ with the four bases of DNA in a macromolecular environment is quite different from the reaction of HO^•^ with simple nucleosides. Indeed, an ambident reactivity has been observed for the reaction of HO^•^ with the G moiety of 2′-deoxyguanosine (2′-dGuo) in aqueous medium [33,34]. The main pathway (~65%) involves H-atom abstraction from the exocyclic NH_2_ group followed by water-assisted tautomerization [35], whereas the minor pathway is the direct addition at the C8 position, resulting in the formation of the adduct radical (17%). H-atom abstraction from the sugar moiety (~18%) completes the picture, half of which occurs at the H5′ positions, followed by radical cyclization with the formation of 5′,8-cyclo-2′-deoxyguanosine [3,36]. On the other hand, the reaction of HO^•^ with 2′-deoxycytidine (2′-dCyd) occurs via addition at C5 and C6 positions in a ca. ratio of 90:10. Experimental evidence regarding H-atom abstraction from amino groups and/or sugar moieties is not available [37], although evidence that the aminyl radical tautomerizes to the most stable forms has been presented [38,39].

The present study provides spectral and kinetic data (obtained via pulse radiolysis) for the various intermediates from the reactions of HO^•^ and SO_4_^•−^ using an alternating GC doubled-stranded 6-mer oligonucleotide, i.e., the ds-ODN of the palindromic 5′-d(GCGCGC)-3′, in deoxygenated aqueous solutions in a condition that excludes the main reactions of HO^•^/O_2_ of Figure 1. This model sequence is also connected with the DNA reactivity, especially with respect to the GC-rich isochores, which are characterized by a high gene density [40,41]. We also theoretically address the reactivity of the G:C complex with HO^•^ by considering all the possible HO^•^ additions and H-atom abstraction from the -NH_2_ group to both G and C moieties, as well as outer-sphere electron transfer from the whole G:C complex to SO_4_^•−^. Moreover, the double-stranded 5′-d(GCGC)-3′ B-DNA sequence was used as a model compound for the theoretical analysis of one-electron-oxidized ds-ODN. Our results are discussed in comparison with other time-resolved spectroscopic studies reported for the reaction of SO_4_^•−^ with similar ds-ODNs [42,43,44,45,46]. We show that interstrand hydrogen bonds can significantly affect the reactivity of HO^•^ with G and C in ds-ODN. Additionally, alternating GC ds-ODN has a remarkable effect on the delocalization of the radical cation and the reaction outcome.

## 2. Materials and Methods

### 2.1. Preparation of Double-Stranded Oligonucleotide (ds-ODN)

The oligonucleotide 5′-d(GCGCGC)-3′ was prepared and purified following protocols described previously [47]. To obtain the ds-ODN, the palindromic strand 5′-d(GCGCGC)-3′ was annealed in buffer solution containing 50 mM sodium phosphate (NaH_2_PO_4_). The substrate was constructed by using a previously reported procedure, showing a melting temperature (T_m_) of 45.9 °C [47]. Analyzing the melting curve at 20 °C, the temperature at which the pulse radiolysis experiments are performed, a percentage of 87% of ds-ODN was estimated (see Appendix A for details).

### 2.2. Pulse Radiolysis

The experiments with time-resolved UV-vis optical absorption detection were carried out at the Institute of Nuclear Chemistry and Technology in Warsaw, Poland. The linear electron accelerator (LAE 10), which delivered 10 ns pulses with an electron energy of about 10 MeV, was applied as a source of irradiation. A detailed description of the experimental setup has been given elsewhere, along with basic details of the equipment and its data collection system [48]. A 150 W xenon arc lamp E7536 (Hamamatsu, Shizuoka, Japan) was used as a monitoring light source. The respective wavelengths were selected by using a MSH 301 (Lot Oriel Gruppe, Darmstadt, Germany) motorized monochromator/spectrograph with two optical output ports. The time-dependent intensity of the analyzing light was measured via means of photomultiplier (PMT) R955 (Hamamatsu, Shizuoka, Japan). A detector signal was digitized using a WaveSurfer 104MXs-B (1 GHz, 10 GS/s, LeCroy) oscilloscope. A water filter was used to eliminate near-IR wavelengths. The optical path of the microcells was 1 cm, with a total volume of irradiated solution of about 300 μL. All experiments were carried out at ambient temperature (~20 °C). The spectral range that can be covered with the existing pulse radiolysis set-up is between 300 and 700 nm. The dosimetry was based on N_2_O-saturated solutions containing 10 mM KSCN, which, following radiolysis, produces (SCN)_2_^•−^ radical anions that have a molar absorption coefficient of 7580 M^−1^ cm^−1^ at *λ* = 472 nm and are produced with a radiation chemical yield of *G* = 0.635 µmol J^−1^ [49]. Absorbed doses per pulse were on the order of about 20 Gy (1 Gy = 1 J kg^−1^).

### 2.3. Computational Details

Equilibrium geometries of the species were evaluated at the density functional level of theory (DFT) using the B3LYP functional with the 6-311++G(d,p) basis set [50]. The Hessian matrix was systematically computed to ascertain the location of the stationary points corresponding to the minimum geometry structures. Solvent (water) effects were included in all computations. The polarizable continuum model (PCM) with standard parameters was used throughout [51]. The unrestricted formalism was adopted for species with unpaired electrons. For reaction products, the wavelengths and oscillator strengths of excited states were predicted via time-dependent (TD) DFT computations by using the same functional and basis set employed for the geometry optimizations.

The starting geometry of the ds-5′-GCGC-3′ DNA sequence was prepared in B-form using 3DNA (version 2.2). Based on calf thymus conformations, the default model of B-DNA was adopted [52]. The negative charge of phosphate units was neutralized by sodium ions [24]. Then, the geometry of [ds-5′-GCGC-3′] ^•+^, i.e., ds-5′-GCGC-3′ oxidized via the removal of a single electron, was optimized at the PCM/DFT level of theory in conjunction with the D3BJ (B3LYP-D3BJ) empirical correction for dispersion energy [53]. The TZVP basis set was used because it has been shown to predict very reliable geometries for short DNA sequences, reaching almost the same quality as more extensive basis sets [54]. The [ds-5′-GCGC-3′] ^•+^ oxidized oligonucleotide was also investigated at the PCM-CAM-B3LYP-D3BJ/TZVP level of computation [55]. After geometry optimizations were carried out at the PCM/B3LYP-D3BJ/TZVP level, TD-B3LYP/TZVP computations were carried out for ^me^G^•+^, the [^me^G:C^me^] ^•+^ Watson and Crick complex, and the [^me^G:C^me^/^me^C^me^G] ^•+^ stack. The latter system was also investigated at the PCM/TD-CAM-B3LYP/TZVP level. All DFT and TDDFT computations were carried out by using the Gaussian package [56].

## 3. Results

### 3.1. Pulse Radiolysis Study

Radiolysis of neutral water leads to the reactive species e_aq_^−^, HO^•^, and H^•^, as shown in Reaction 1. The values in parentheses represent the radiation chemical yields (*G*) in units of μmol J^−1^. In N_2_O-saturated solution (∼0.02 M of N_2_O), e_aq_^−^ are converted into HO^•^ radicals via Reaction 2 (*k*_2_ = 9.1 × 10^9^ M^−1^ s^−1^), with *G*(HO^•^) = 0.56 μmol J^−1^ [57,58].
H_2_O + γ-irr. → e_aq_^−^ (0.28), HO^•^ (0.28), H^•^ (0.06)(1)
e_aq_^−^ + N_2_O + H_2_O → HO^•^ + N_2_ + HO^−^(2)
HO^•^ + ds-ODN → radical products(3)

The reaction of HO^•^ with the palindromic 5′-d(GCGCGC)-3′ doubled-stranded oligonucleotide (ds-ODN) was investigated in phosphate-buffered (50 mM) N_2_O-saturated solutions of 0.134 mM ds-ODN at natural pH (pH 7 was recorded). Transient absorption spectra in the range of 300–700 nm recorded at 4.8, 60, and 1000 μs after the electron pulse are presented in Figure 2. The transient spectrum obtained at 4.8 μs shows a dominant absorption band (λ = 310 nm) that constantly decreases up to 700 nm with a weak shoulder in the 400–450 nm range. The spectrum recorded at 60 µs is slightly less intensive and did not show any relevant change in the 300–700 nm range. With the further elapse of time, the spectrum observed at 1000 µs exhibited a strong band with no distinct λ_max_ at wavelengths in the range of 300–700 nm. A rate constant of 1.4 × 10^10^ M^−1^ s^−1^ was determined for Reaction 3 by measuring the pseudo-first-order rate constant of the increase in the absorbance at λ_max_ = 310 nm for the specified concentration of ds-ODN (0.134 mM) and assigned to the main transient formed (the top inset of Figure 2). In turn, its disappearance followed first-order kinetics, with k = 1.6 × 10^3^ s^−1^ (see the bottom inset of Figure 2).

The sulfate radical anion SO_4_^•−^ was generated by irradiating an Ar-purged solution containing 20 mM of ammonium persulfate in the presence of 0.1 M of tert-butanol; under these conditions, e_aq_^−^ and H^•^ are converted into SO_4_^•−^ via Reaction 4 (*k*_4_ = 1.2 × 10^10^ M^−1^ s^−1^) and Reaction 5 (*k*_5_ = 2.5 × 10^7^ M^−1^ s^−1^), respectively, whereas HO^•^ radicals are scavenged by the alcohol (Reaction 6) [57].
e_aq_^−^ + S_2_O_8_^2−^ → SO_4_^•−^ + SO_4_^2−^(4)
H^•^ + S_2_O_8_^2−^ → H^+^ + SO_4_^•−^ + SO_4_^2−^(5)
HO^•^ + (CH_3_)_3_COH → H_2_O + ^•^CH_2_(CH_3_)_2_COH(6)
SO_4_^•−^ + ds-ODN → radical products(7)

Figure 3 shows the optical absorption spectra of SO_4_^•−^ (black circles) and the new transient spectrum obtained from the reaction with ds-ODN (red circles) (taken at 32 ns and 800 ns after the electron pulse, respectively). The rate constant of 8.2 × 10^10^ M^−1^ s^−1^ was determined for the reactions of SO_4_^•−^ with ds-ODN at pH 7 (Reaction 7) by measuring the growth of absorbance of a new transient species at λ_max_ = 330 nm (the top inset of Figure 3). Furthermore, the disappearance of this species gives rise to another transient, the spectrum of which (taken at 60 µs after the pulse) is also shown in Figure 3 (green circles). A first-order rate constant of 1.5 × 10^5^ s^−1^ was measured for the transformation of the red spectrum to the green one by following the rate of the decrease in absorbance at λ = 330 nm (see the bottom inset of Figure 3).

Figure 4 presents the optical absorption spectra of the two transients obtained by reactions of HO^•^ (red circles) and SO_4_^•−^ (black circles) with ds-ODN taken at 1000 µs after the pulse, taking into consideration the radiation chemical yields (*G*) of the two reactive species. It is gratifying to see that the two spectra are nearly identical in terms of feature and intensity. Moreover, the disappearance of both transients observed at λ = 310 nm followed first-order kinetics, with the two having similar first-order rate constants of 1.6 × 10^3^ s^−1^ and 1.1 × 10^3^ s^−1^ for the reactions of HO^•^ (red circles) and SO_4_^•−^ (black circles), respectively (see insets in Figure 4).

### 3.2. Theoretical Calculations

The reactivity of the Watson–Crick G:C complex with HO^•^ has been previously considered [59,60,61]. HO^•^ addition to the C4, C5, and C8 carbon sites of G the moiety and to the C5 and C6 carbons of the C moiety (see Figure 5 for atom numbering, where sugar-phosphate moieties have been replaced by methyl groups (viz., ^me^G:C^me^)) leads to stable adducts with respect to possible proton transfer processes within the G:C pair. The addition of HO^•^ to the C8 of ^me^G has been predicted to be the most favored process. Previous computations have also predicted that the addition of HO^•^ to the C5 and C6 sites of ^me^C are more favored than those to the C4 and C5 of ^me^G. This has been traced back to the significant distortion from planarity of the whole G ring upon the addition of HO^•^ to the C4 and C5 sites of G [59]. Our own computations confirmed all of these results, cf. Table 1 for the predicted energy changes (Δ*E*, kcal/mol); furthermore, no barriers were predicted for any of the HO^•^ addition reactions. A desolvation barrier could be expected since HO^•^ is significantly solvated in water [62,63], but it could hardly discriminate among the different sites of the G:C complex; thus, it was not expected to affect the selectivity of HO^•^ addition toward the different sites of the G:C complex.

We also considered H-atom abstraction by HO^•^ from the -NH_2_ group of both G and C. As expected [33,34], H-atom abstraction from the exocyclic amino group of ^me^G was predicted to be significantly more favored than that from ^me^C (Table 1, columns 4 and 7). No barriers have been found for both processes. H-atom abstraction from G takes place even when HO^•^ is completely surrounded by the water molecules of its first solvation shell via the network of H-bonds connecting the solvated HO^•^ to the amino group of G. Even though H-atom abstraction from the G moiety is less exergonic than HO^•^ addition to C8, the former process is expected to be entropically favored in water solution. Evaluating reliable entropic changes in solution at this level of computation is not possible, and calculating Δ*G* is a highly demanding computational task [64]. Here, it is important to note that the large internal energy gain (ca. 10 kcal/mol), which favors HO^•^ addition to C8 with respect to H-atom abstraction, can be much smaller when Gibbs free energies are considered.

The spectroscopic features of all the HO^•^ adducts were predicted by using time-dependent density functional theory (TDDFT); the absorption wavelengths (*λ*) and oscillator strengths (*f*) of the most intense transitions are reported in Table 1. All the HO^•^ adducts exhibit absorptions in the visible region. In particular, the TDDFT computations predicted for the C8 adduct show two intense absorptions at the highest energy region of the visible spectrum—430 and 397 nm with *f* of 0.019 and 0.021, respectively—and three much more intense absorptions at 376, 325, and 309 nm, with *f* of 0.040, 0.041 and 0.073, respectively (Table 1, column 1). The situation is somehow different for the HO^•^ adduct formed at the C6 carbon of the cytosine moiety (the second most stable adduct). In that case, TDDFT predicted a very weak absorption at 808 nm and a more intense one at 632 nm, with *f* = 0.015 (Table 1, column 6); those absorptions occurring at longer wavelengths have no counterpart for any other HO^•^ adduct. The most intense electronic transitions predicted by the TDDFT computations are those for the neutral aminyl radicals of both the G and C moieties (Table 1, columns 4 and 7). For G(-H):C, two relevant transitions are predicted in the visible region: a comparatively stronger absorption at 567 nm (*f* = 0.079) and a weaker one at 447 nm (*f* = 0.011). The most intense transition falls in the UV region, being predicted at 309 nm, with *f* = 0.085. The predicted spectrum agrees quite well with that obtained via the pulse radiolysis of the G water solution at neutral pH. For the GC(-H) complex, the predicted absorption spectrum is very different, exhibiting a very weak transition in the NIR region (at 1788 nm), a second, more intense transition at 874 nm (*f* = 0.033), and an intense transition at 338 nm (*f* = 0.028).

We have also considered the addition of SO_4_^•−^ to the C8 site of ^me^G and to the C5 and C6 of ^me^C, as well as the spectroscopic features of the adducts. The results, which are summarized in Appendix A of the Appendix A, indicate much less exergonic reactions.

Although HO^•^ radicals can oxidize DNA nucleobases via inner-sphere electron transfer (ET), which occurs via radical addition to a carbon atom of the G:C complex followed by HO^−^ elimination, the SO_4_^•−^ is known to act via outer-sphere ET. Initially, we considered a model in which the first hydration shell of SO_4_^•−^ and its ET products were explicitly included in the computation. The optimized structure for SO_4_^•−^ is shown in Appendix A of the Appendix A. We assumed a tetrahedral arrangement of water molecules around each oxygen atom, considering twelve water molecules for SO_4_^•−^, without attempting to verify whether larger models could be more appropriate. Bulk solvent effects were considered by the polarizable continuum model (PCM) model. As expected, with the inclusion of specific interactions with solvent molecules, ET is predicted to be significantly exergonic for SO_4_^•−^ (Δ*E* = −26.7 kcal/mol). The above energy change for the outer-sphere ET reaction is in line with those inferred from the standard redox potentials of SO_4_^−^/SO_4_^2−^ and guanosine 5′-monophosphate (GMP) GMP+/GMP (pH 7) semi-couples, which, according to cyclic voltammetry, are 2.60 and 1.31 V (vs. SHE), respectively [65,66]. In summary, outer-sphere ET from the G:C complex to SO_4_^•−^ is predicted to be the most favored process, being largely more exergonic than the direct addition to both the G and C moieties.

Regarding the spectroscopic properties of the [G:C]^•+^, they have been reported previously [67,68]: a long wavelength absorption is predicted in the NIR region of the spectrum at 11,218 cm^−1^ (*f* = 0.006) and observed at 10,200 cm^−1^ [67], together with two more intense absorptions at 528 nm (*f* = 0.019) and at 370 nm *(f* = 0.081). However, in the case of the present study, the UV absorption spectrum could significantly differ from that of a single G:C pair because the radical cation of ds-ODN of the palindromic 5′-d(GCGCGC)-3′ sequence could, in principle, be delocalized over the whole double strand or over a part (the central one) of it. Delocalized hole domains have been experimentally found both in single and double strands containing sequences of consecutive adenines [25,69] and theoretically predicted by DFT computations [24,70]. In the case of G-rich ODN, the situation is less clear inasmuch as differential pulse voltammetries of guanine-rich oligonucleotides containing up to six consecutive guanines showed a progressive decrease in the first voltammetric peak potential as the number of adjacent guanines increased, indicating the establishment of delocalized hole domains [25], whereas ab-initio and DFT computations including solvent effects predict that the hole is localized over a single G, the one located at the 5′ side [71,72].

The double-stranded 5′-d(GCGC)-3′ palindromic B-DNA sequence was used as a model compound for the theoretical analysis of one-electron oxidized ds-ODN. The predicted (B3LYP) spin density of this oxidized model is depicted in Figure 6. Both B3LYP and CAM-B3LYP (see Appendix A) functionals find the positive charge almost equally distributed over the central guanines. Indeed, Mulliken spin densities of central guanines are predicted to be 0.50, 0.48 by B3LYP and 0.48, 0.52 by CAM-B3LYP. Therefore, computations suggest that states with the hole fully localized on a single guanine are sufficiently coupled with each other to overcome solvent effects that favor charge localization [72,73]. Such a result is remarkable because, in single-stranded DNA oligomers ss-5′-XGGY-3′ (X,Y = C,T,A, and 6-azauracil), the electron hole is never predicted to be delocalized on the two stacked guanines [72].

In order to make TDDFT computations feasible, sugar phosphate moieties were replaced by methyl groups (viz., ^me^G:C^me^). A partial geometry optimization was carried out at the PCM/B3LYP-D3/TZVP level by starting from the optimized geometry of ds-GCGC and only relaxing the atoms of the methyl groups. The lowest energy electronic transition of the alternating GC pairs [^me^G:C^me^/^me^C:G^me^]^•+^ was predicted by B3LYP computations to occur at 1839 nm, i.e., ∆*E* = 0.67 eV. This HOMO–LUMO intense absorption involves π MOs with electron densities equally delocalized over the guanines. By using a simple two-state model, such an absorption energy would yield an electronic coupling term between the fully localized diabatic states roughly amounting to 0.33 eV, suggesting that the geometrical arrangement of guanines is very effective in stabilizing the positive charge in oxidized B-DNA sequences composed by alternating GC pairs. CAM-B3LYP (see Appendix A and Appendix A) also finds an intense lowest energy transition involving MOs with charge equally distributed on guanines. However, a substantially lower absorption energy ∆*E* = 0.20 eV is predicted by the latter functional. A spectroscopic analysis of oxidized ds-ODN in the near- and medium-IR spectral region would be highly advisable to ascertain the effectiveness of interacting guanines in stabilizing the hole in the alternating GC sequences.

The predicted spectrum of [^me^G:C^me^/^me^C:G^me^]^•+^ is reported in Figure 7 (red line), together with the signal of Figure 3, which was recorded at 800 ns after the pulse. Apart from the intense absorption in the region of the longer wavelength, attributed to the HOMO–LUMO transition, with both levels delocalized over the central guanines, the predicted spectrum exhibits intense absorptions only at significantly shorter wavelengths, i.e., below 270 nm. All the computed transitions are reported in the Appendix A, together with the results of previous computations concerning the single radical cation ^me^G^•+^ and the Watson–Crick [^me^G:C^me^]^•+^ complex, whose observed spectra have been reported in the literature, thus being extremely useful for assessing the accuracy of the predicted transition energies. The comparison between the observed and calculated spectra of those latter species shows that the TDDFT computations largely overestimate excitation energies at shorter wavelengths, as also reported by other authors [46,68]. Notwithstanding, the TDDFT computations correctly predict a blue shift of the most intense signal and an increase in the intensity of the most intense absorption with respect to signals at longer wavelengths. Although most of the predicted transitions for [^me^G:C^me^]^•+^ involve the π system of guanine, the higher intensity observed for [Guo:dCyd]^•+^ with respect to Guo^•+^ in the region around 300 nm is ascribed by the TDDFT computations to π→π* transitions that also involve cytosine. This effect is dramatically enhanced in oxidized alternating GC sequences. Indeed, the predicted spectrum of [^me^G:C^me^/^me^G:C^me^]^•+^ is characterized by a very high ratio between the intensity at 260 nm and the one at 375 nm with respect to the spectra of [^me^G:C^me^]^•+^ or ^me^G^+•^. That high intensity is ascribed by the TDDFT computations to the strong coupling between oxidized guanines, which gives rise to several intense absorptions between 260 and 275 nm, involving the G/G π system. Furthermore, the intense π→π* absorptions of the C/C subsystem, together with charge transfer transitions, moving the hole from the G/G to the C/C units, and transitions that localize the hole on a single G moiety, are predicted in the highest energy region, making the absorption signal even more intense.

## 4. Discussion

The rate constants for the reactions of HO^•^ with 2′-deoxyguanosine (2′-dGuo) and 2′-deoxycytidine (2′-dCyd) were found to be similar (Table 2). We measured the rate constant of HO^•^ with the alternated GC ds-ODN to be 1.4 × 10^10^ M^−1^ s^−1^. Assuming that the six G:C pairs in ds-ODN of the palindromic 5′-d(GCGCGC)-3′ have similar reactivities, we calculated a value of 2.3 × 10^9^ M^−1^ s^−1^ for the reaction of HO^•^ with the G:C pair, which is approximately five-fold smaller than the sum of the two single nucleoside values participating in the pair. On the other hand, taking into account the concentration of the phosphate buffer (50 mM) used in our study, this rate constant is nearly three-fold larger than the value measured for the ct-DNA [74].

The rate constant for the reactions of SO_4_*^•−^* with 2′-dGuo is reported to be 2.5-fold faster than that with 2′-dCyd (Table 2). We measured the rate constant of SO_4_*^•−^* with the alternated GC ds-ODN to be 8.2 × 10^10^ M^−1^ s^−1^. Assuming that the six G:C pairs in ds-ODN have similar reactivities, we calculated a value of 13.6 × 10^9^ M^−1^ s^−1^ for the reaction of SO_4_^•−^ with the G:C pair, which is 3.3- and 8.5-fold larger than the value for the single nucleoside 2′-dGuo and 2′-dCyd and ~2.5-fold larger than the sum of the two single nucleoside values participating in the pair. Interestingly, this value is very close to the rate constants of the reactions of SO_4_^•−^ with the 30mer ODN containing G moieties in ss-ODNs and ds-ODN [43].

Based on the present study’s time-resolved spectroscopy results and theoretical calculations, we assigned the observed spectrum at 4.8 μs for the reaction of HO^•^ with ds-ODN (Figure 2), i.e., the absorption band with λ = 310 nm that constantly decreases up to 700 nm, to HO^•^ radical adduct [8-HOG:C]^•^ (cf. Figure 8). A clean UV-vis spectrum of the HO^•^-adduct to the C8 position of guanine derivatives, i.e., 8-HO-G^•^, is unknown. An indirect experiment (resulting from the spectra subtraction) and theoretical calculation indicated an intense band near 300 nm, with a broad shoulder up to 500 nm, containing a less intense band between 400 and 450 nm [33]. This description fits well with the recorded spectrum at 4.8 μs in Figure 2. However, we cannot exclude the notion that the weakly pronounced shoulder in the 400–450 nm range may be also due to HO^•^ adducts either at G5 and/or C5, based on the predicted absorption wavelengths and oscillator strengths reported in Table 1. Interestingly, the broad absorption band with λ_max_ ≈ 600 nm was not observed at short times, suggesting that HO^•^ radicals do not react with ds-ODN via H-atom abstraction from the exocyclic NH_2_ group in the guanine (G) moiety but rather via addition to the C8 position [30]. However, the observed spectrum matches quite nicely with the transient spectra resulting from the reaction of HO^•^ radicals with ct-DNA [74] and 12-mer ds-ODN [75]. The absence of H-atom abstraction from the exocyclic NH_2_ group in the guanine (G) moiety might result from the specific structure of ds-ODN, which does not allow HO^•^ radicals to reach the NH_2_ moiety [76], as also suggested for 12-mer ds-ODN [75]. A similar though slightly less intense spectrum was recorded at 60 µs, indicating no trace of H-atom abstraction from the C5′ position with subsequent radical cyclization to the C8 position in our pulse radiolysis study, as has been previously noted [75]; however, this certainly exists in the background, as demonstrated by numerous product studies [8]. Indeed, this sequence of reactions in naked DNA or ds-ODN is expected to account for the 1–2% of the whole HO^•^ reactivity.

On the other hand, the reaction of SO_4_*^•−^* with ds-ODN, reported in Figure 3, can simply be rationalized as follows: one-electron oxidation of ds-ODN followed by the hydration of [G:C]^•+^ to give [8-HOG:C]^•^ with a pseudo-first-order rate constant of *k*(H_2_O) = 1.5 × 10^5^ s^−1^ (Figure 8). Indeed, Figure 4 shows that the absorption spectra of both HO^•^ and SO_4_^•−^ with ds-ODN (taken at 1000 µs after the pulse) are similar in terms of features and intensity, taking into consideration the radiation chemical yields (*G*) of the two reactive species. Moreover, the kinetics of disappearance of both transients followed similar first-order kinetics, which we tentatively assigned to the ring opening of [8-HOG:C]^•^ with the formation of [Fapy-G:C]^•^ (Figure 8). It is worth mentioning that the reported rate constants regarding the water trapping of G^•+^ in ds-ODN are only estimated values [30]. The value of *k*(H_2_O) = 6 × 10^4^ s^−1^ was calculated for the guanine radical cation in the GGG sequence via fitting using known charge transfer rates and the product yields at pH 7. The water trapping rates vary substantially, with G^•+^ being more reactive than GG^•+^ and pH-dependent. All together, our experimental values regarding water trapping fit very well with these considerations [30]. It is worth mentioning that the G^•+^ in the single-stranded ODN 5′-d(TCGCT) at pH 2.5, where the transient exists in the form of a radical cation, decays with a pseudo-first-order rate constant of *k*(H_2_O) = 3.3 × 10^2^ s^−1^ [76].

The above proposal requires further consideration with respect to the spectrum shape of one-electron oxidized ds-ODN and the role of the well-known deprotonation of [G:C]^•+^ in the case of the palindromic 5′-d(GCGCGC)-3′. As mentioned in the introduction, the primary damage in one-electron-oxidizing DNA is localized at Gs, which have the lowest oxidation potential. Since the earliest works in this field, it has been proposed that, in the [G:C]^•+^ pair, the proton is not directly lost from G^•+^ to the aqueous phase but remains within the hydrogen-bonded interaction with the cytosine N3 atom [77]. ESR studies on a variety of ds-ODNs have indicated that a prototropic equilibrium [G:C]^•+^ ⇆ [G(-H^+^):C(+H^+^)]^•+^ should be established at ambient temperature (Figure 8) and that the proton is entirely transferred at 77 K [26]. The reaction of SO_4_^•−^ of a variety of ds-ODNs containing G, GG, and GGG sequences has been reported in pulse radiolysis studies [42,78]. In an earlier work [78], the biphasic decay of [G:C]^•+^ was attributed to the shift of a proton from the N1 in G^+•^ to the N3 of C, followed by the release of the proton into the solution. A few years later, the same group analyzed the optical spectra and the kinetics of eleven ds-ODNs with different sequences at 625 nm (11mer to 13mer ds-ODNs) and proposed that the monophasic decay of [G:C]^•+^ ⇆ [G(-H^+^):C(+H^+^)]^•+^ is associated with the release of the proton into the solution [42]. The rate constant for deprotonation was dependent on the ds-ODN sequence, varying in the range of 0.3–2 × 10^7^ s^−1^, as observed by monitoring the transient at 625 nm. One of the sequences particularly relevant to the present work contains alternating GC sequences, the 11-mer 5′-d(CGCGCGCGCGC)-3′ and its complementary strand, which have been associated with a faster rate constant of 2 × 10^7^ s^−1^. Unfortunately, no data are shown for this ds-ODN. In our experiment (see Figure 3, red spectrum), the transient absorbance at 625 nm is close to zero, whereas we attribute the decay at 330 nm with *k* = 1.5 × 10^5^ s^−1^ to the hydration of [G:C]^•+^. In this context, it is also worth mentioning that our theoretical results indicate the delocalization of positive charge over the central guanines on the two stacked base pairs (see Figure 6), suggesting that the *K*_eq_ in Figure 8 may be shifted strongly to the left since the positive charge in the palindromic 5′-d(GCGCGC)-3′ sequence may be delocalized over the whole ds sequence or part (the central one) of it.

The one-electron oxidation of ds-ODN of 30-mer 5′-CGTACTCTTTGGTGGGTCGGTTCTTTCTAT-3′ (G, GG, and GGG sequences) and its complementary strand by SO_4_^•−^ was studied via laser flash photolysis by the authors of [43]. In this study, the SO_4_^•−^ was produced via the photolysis of S_2_O_8_^2−^ using 308 nm laser pulses, and the transient species was observed after the complete decay of SO_4_^•−^ (5 μs), which was assigned to G(N1-H)^•^ and characterized by a narrow absorption band at 312 nm and two less intense bands near 390 and 510 nm [30,43]. The decay of G(N1-H)^•^ was biphasic, with one component decaying with a lifetime of ∼2.2 ms and the other one with a lifetime of ∼0.18 s. The authors proposed that the equilibrium [G:C]^•+^ ⇆ [G(-H^+^):C(+H^+^)]^•+^ allows for the hydration and subsequent formation of 8-oxoG:C. The same ds-ODN were studied using photoionization via the direct absorption of low-energy photons, and the spectra at 5 µs are the same as above [45].

Our spectrum regarding the one-electron-oxidized palindromic 5′-d(GCGCGC)-3′ sequence registered at 800 ns (Figure 3) shows a completely different shape of the G(N1-H)^•^ neutral radical. Based on the described calculations, we assigned the species to [G:C]^•+^ that may be stabilized by delocalization over the whole double strand or over a part (the central one) of it. Indeed, the calculated spectral shape agree very well with the experimental one, cf. Figure 7. The fact that the absorption peak is predicted at much shorter wavelengths than the observed one (ca. 70 nm) is a well-known failure of TDDFT, irrespective of the adopted functional [46,67]. Here, the problem is further exacerbated by the large size of the system considered in computations (see Figure 6), but with sugar phosphate moieties replaced by methyl groups. Indeed, the most intense transition predicted at ca. 260 nm corresponded to the 56th excited state, and our TDDFT computations considered only 65 excited states; therefore, the computed transition energies are highly inaccurate at shorter wavelengths. Interestingly, the alternating GC sequence in duplexes has been previously studied using different techniques. The transient species generated via the direct absorption of the low-energy UV irradiation of ds-ODN palindromic 5′-GCGCGCGCGC-3′ and register at 100 μs was assigned to G(N1-H)^•^ [44]. This spectrum is very similar to our green spectrum in Figure 3, which we assigned to the HO^•^-adduct, i.e., [8-oxo-G:C]^•^.

## 5. Conclusions

In this work, we evaluated the reactivities of HO^•^ and SO_4_^•−^ with respect to the alternating GC double-stranded 5′-d(GCGCGC)-3′ using time-resolved spectroscopy at a nanosecond timescale. The assignment of transient species was corroborated by a tailored computational study. A reaction with SO_4_^•−^ results in the formation of the ds-ODN^•+^, where the electron hole is delocalized over the whole or part of the alternating GC sequence. This is a new type of electron hole stabilization in CG-rich sequences of DNA. Indeed, the ODN^•+^ prefers the reaction with water (*k* = 1.5 × 10^5^ s^−1^) instead of the deprotonation usually observed in other studies. The neutral radical [8-oxo-G:C]^•^ is a common transient of both oxidizing species (Figure 8). The biological implications of the proposed doubled-stranded GC alternating sequences regarding hole delocalization in DNA pertain to the generation of guanine damage.

## Figures and Tables

**Figure 1 biomolecules-13-01493-f001:**
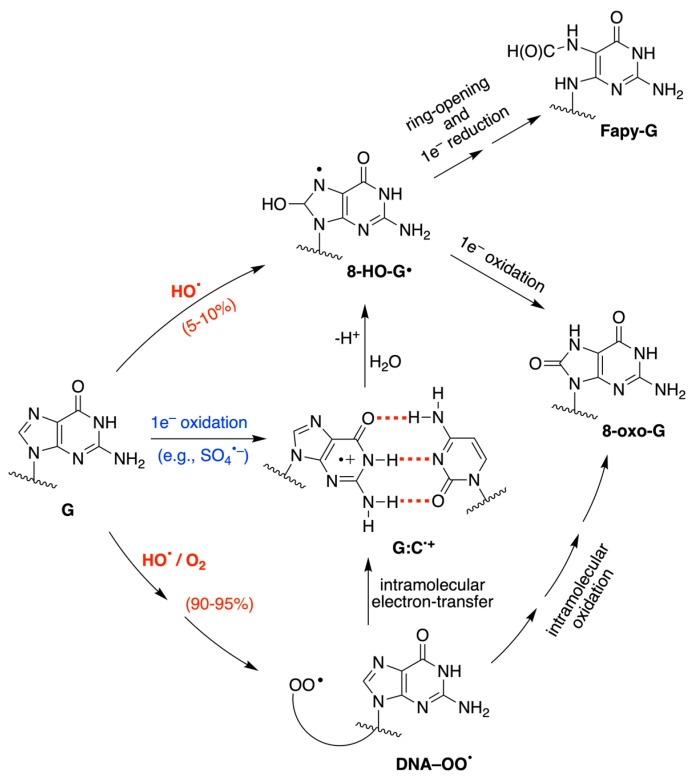
Proposed mechanism for the reactions of HO^•^ and SO_4_^•−^ with G moieties in ds-ODNs and DNA; the reactions lead to the formation of 8-oxo-G and Fapy-G as stable lesions [6,7,8].

**Figure 2 biomolecules-13-01493-f002:**
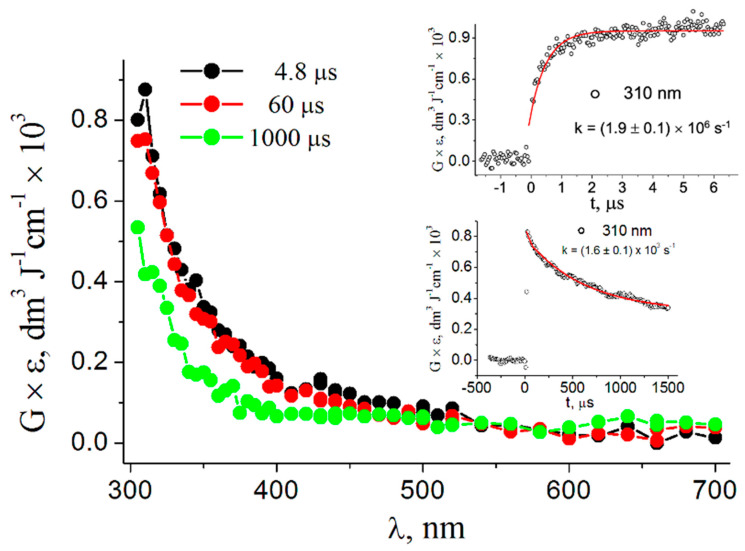
Absorption spectra obtained from the pulse radiolysis of a N_2_O-saturated phosphate-buffered (50 mM) solution containing 0.134 mM ds-ODN at pH 7 (taken at 4.8, 60, and 1000 µs after the electron pulse). Inset: Time dependence of absorption at 310 nm (build-up and decay) recorded in the same experimental conditions of the spectra; the red lines represent the first-order kinetic fit to the data.

**Figure 3 biomolecules-13-01493-f003:**
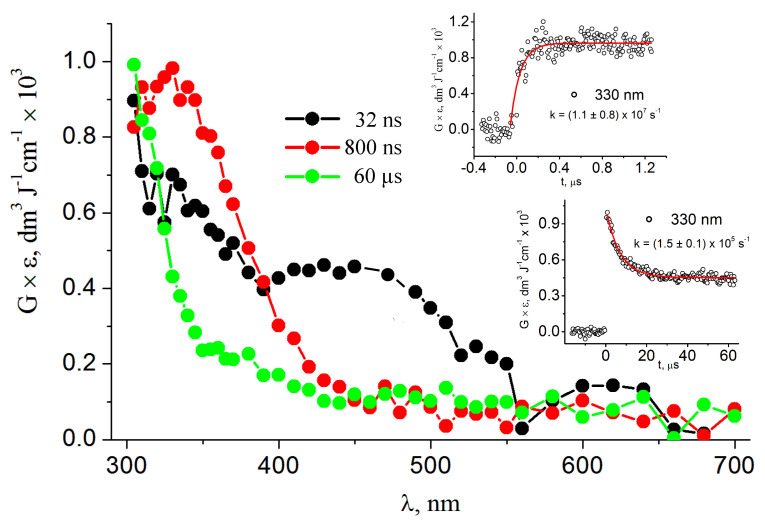
Absorption spectra obtained from the pulse radiolysis of an Ar-purged phosphate-buffered (50 mM) solution containing 0.134 mM ds-ODN, 20.0 mM ammonium persulfate, and 0.1 M tert-butanol at pH 7 (taken at 32 ns, 800 ns, and 60 µs after the pulse). Inset: Time dependence of absorbance at 330 nm (build-up and decay) (recorded in the same experimental conditions of the spectra); the red lines represent the first-order kinetic fit to the data.

**Figure 4 biomolecules-13-01493-f004:**
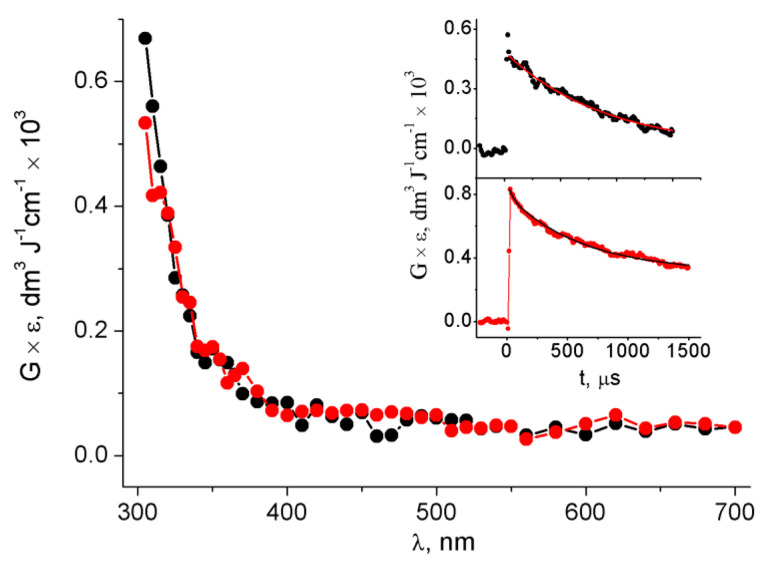
Comparison of the absorption spectra obtained via pulse radiolysis from the reaction of 0.134 mM ds-ODN with HO^•^ (red circles) and SO_4_^•−^ (black circles) in phosphate-buffered (50 mM) solution at pH 7 (taken at 1000 µs after the pulse). Insets: Time dependence of absorbance at λ = 310 nm, which was recorded in the same experimental conditions as the transient spectra obtained from the reaction involving SO_4_^•−^ (top inset) and HO^•^ (bottom inset) radicals.

**Figure 5 biomolecules-13-01493-f005:**
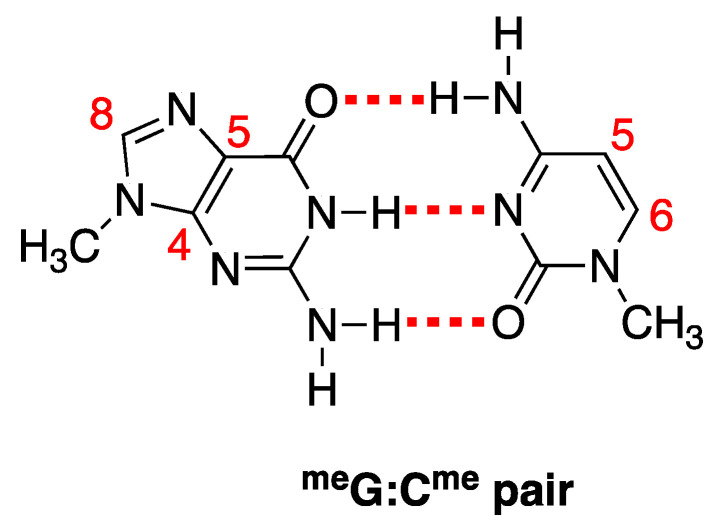
Adopted atom numbering of the ^me^G:C^me^ pair.

**Figure 6 biomolecules-13-01493-f006:**
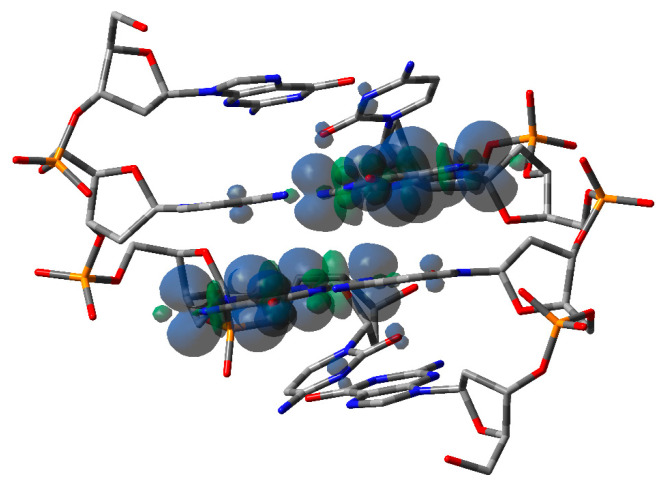
Optimized geometry and spin density plot (PCM/B3LYP-D3BJ/TZVP) of [ds-5′-GCGC-3′]^•+^. Hydrogen atoms and sodium counterions have been omitted for clarity.

**Figure 7 biomolecules-13-01493-f007:**
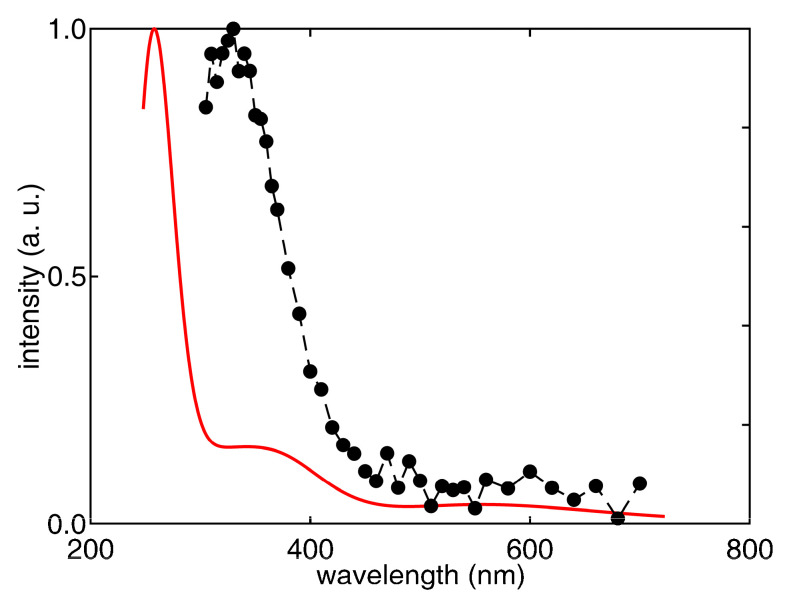
Predicted (PCM-TD-B3LYP/TZVP) UV/vis spectrum of [^me^G:C^me^/^me^C:G^me^]^+•^ (red line); vertical transitions have been enlarged by Gaussian functions with full width at half-maximum, amounting to 6000 cm^−1^, and the heights have been scaled according to the predicted oscillator strengths of Appendix A in the Appendix A. The absorption spectra obtained from the pulse radiolysis of Ar-purged ds-ODN with ammonium persulfate (taken at 800 ns after the exciting pulse) are also shown for comparison (black curve).

**Figure 8 biomolecules-13-01493-f008:**
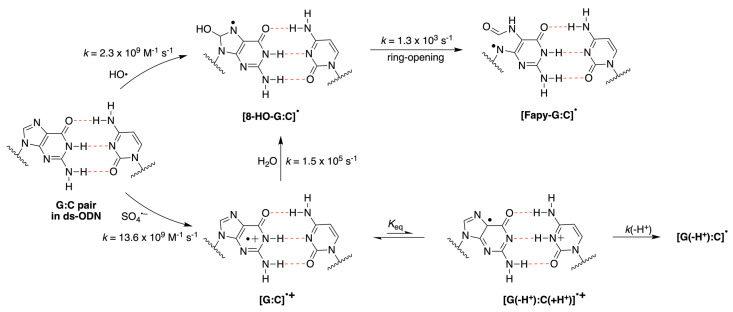
Proposed reaction mechanism for the reaction of HO^•^ and SO_4_*^•−^* with the ds-ODN of the palindromic 5′-d(GCGCGC)-3′ in deoxygenated aqueous solutions. The reported rate constants for the reactions of HO^•^ and SO_4_^•−^ are calculated for a G:C pair in alternating GC oligonucleotides.

**Table 1 biomolecules-13-01493-t001:** Predicted energy changes (Δ*E*, kcal/mol) for the addition of HO^•^ to ^me^G:C^me^ pairs in water and for H-atom abstraction from the NH_2_ moieties of G or C by HO^•^; the absorption wavelengths (*λ*, nm) and oscillator strengths (*f*) of all the HO^•^ adducts in water are also shown.

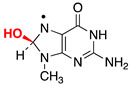	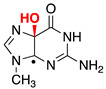	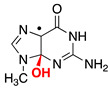	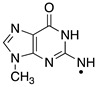	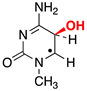	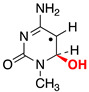	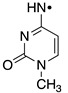
Δ*E* = −31.7	Δ*E* = −9.9	Δ*E* = −15.0	Δ*E* = −21.7	Δ*E* = −17.3	Δ*E* = −18.6	Δ*E* = −9.4
*λ*, nm	*f*	*λ*, nm	*f*	*λ*, nm	*f*	*λ*, nm	*f*	*λ*, nm	*f*	*λ*, nm	*f*	*λ*, nm	*f*
430	0.019	535	0.008	481	0.007	595	0.006	420	0.065	808	0.003	1788	0.003
397	0.021	483	0.074	437	0.009	570	0.001	346	0.024	632	0.015	874	0.033
386	0.002	402	0.033	399	0.005	567	0.079	313	0.002	488	0.021	338	0.028
376	0.040	381	0.006	377	0.014	482	0.007	312	0.028	413	0.002	270	0.011
353	0.002	363	0.036	362	0.054	447	0.011			373	0.001	265	0.108
339	0.010	360	0.053	356	0.017	350	0.056			368	0.001		
325	0.041	340	0.005	350	0.059	309	0.085			324			
309	0.073	328	0.003	347	0.038								
308	0.019	321	0.011	337	0.034								

**Table 2 biomolecules-13-01493-t002:** Rate constants for the reactions of HO^•^ and SO_4_^•−^ with 2′-dGuo, 2′-dCyd, and the G:C pair in ds-ODN.

Substrate	*k*(HO^•^), M^−1^ s^−1^	*k*(SO_4_^•−^), M^−1^ s^−1^
2′-dGuo ^1^	5.7 × 10^9^	4.1 × 10^9^
2′-dCyd ^1^	5.6 × 10^9^	1.6 × 10^9^
G:C pair in ds-ODN ^2^	2.3 × 10^9^	13.6 × 10^9^

^1^ Taken from [5]. ^2^ Present work.

## Data Availability

The data presented in this study are available in this article and the Appendix A.

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
