# Peer review of "Hydroxyl Radical vs. One-Electron Oxidation Reactivities in an Alternating GC Double-Stranded Oligonucleotide: A New Type Electron Hole Stabilization"

_biomolecules, 2023, doi:10.3390/biom13101493_

Round 1

Reviewer 1 Report

This interesting manuscript investigates the reactivity of hydroxyl radical and sulfate anion radical toward a double stranded oligonucleotide containing -GC- tracks. The same transient species corresponding the [8-OH-G:C]dot adduct is observed in both cases. The proposed mechanisms are rationalized by computational studies predicting the energy changes for addition of hydroxyl radical or sulfate radical anion at different G or C sites. The experimental absorption spectrum of the radical is also in good agreement, in shape, with the theoretical one, and point toward a new type of electron hole stabilization in alternating GC sequences. This work deserves publication in Biomolecules after addressing the two following points.

-          The final conclusion of this work is the proposal of an electron hole stabilization. In this context, hybridization is an important issue as experiments were performed with a short double stranded oligonucleotide containing only 6 base pairs. More details are needed to assess the correct formation and stability of the duplex.

-          The rate constant for reaction between sulfate anion radical and the ds-ODN, reaction (7), is determined from the growth at 330 nm of the new transient. It should be interesting to confirm this value by considering the decay of the sulfate radical anion at ca. 450 nm.

Reviewer 2 Report

Understanding the underlying molecular reaction mechanism of ROS with DNA is of great importance. In this work, the authors presented spectral and kinetic data (ns - ms) for the reactions of HO· and SO4·- with an alternating GC double-stranded ODN. Combined with theoretical studies, the reactivities of G:C complex with HO·/ SO4·- (addition, H-atom abstraction, and electron transfer) are addressed. Moreover, it is interesting that the authors assigned the ds-(GC)6 transient spectrum at 800 ns as [G:C/C:C]·+ in which the electron hole is predicted to be delocalized on the two stacked base pairs. Overall, this work broadens our standing of the reaction damage mechanism of DNA by ROS. These data appear reliable and the analysis perform fine. I recommend the acceptance of the manuscript for publication, provided that the authors manage to address the following issues.

1.     Page 5:187-189 The authors assigned the increase of the absorbance at 310 nm as the reaction 3 process. Could the authors give some explanations to exclude the possible assignment of this increase as the subsequent reaction process (e.g. deprotonation) rather than the reaction 3 process? Maybe the reaction 3 is too fast to be observed?

2.     Besides, after carefully examining the fit line and the kinetic data, the increase of the absorbance at 310 nm seems to follow a bi-exponential behavior rather than mono-exponential behavior.

3.     Page5:209-211. Similar questions. Please see comment 1 and 2.

4.     For the 3.2 Theoretical Calculations, do the authors consider the effects of explicit water molecules when optimizing the model structure? In previous works, the role of explicit water molecules has been well addressed. (e.g. J. Phys. Chem. B 2006, 110, 2417124180; J. Phys. Chem. B 2009, 113,1135911361; J. Phys. Chem. B 2010, 114, 1343913445;)

5.     What if the alternating GC double-stranded ODN (ds-(GC)6) is replaced by ds-(GGGCCC) or ds-(GGCCGG/CCGGCC)? Any difference for the [G:C/C:C]·+?

Reviewer 3 Report

The manuscript from Masi et. al. compares the reactivity of hydroxyl radical and sulfate radical anion with the GC double-stranded oligonucleotide. The manuscript is well written and the authors claim to have identified a new mode of electron-hole stabilization. I have the following comments/questions before I can recommend this manuscript for publication. 

·        Authors in a previous paper ‘Chem. Res. Toxicol. 2011, 24, 2200–2206’ have demonstrated that hydroxy radical under anoxic conditions with 2’deoxy guanosine can generate a 5’-sugar radical by H atom abstraction which leads to a cyclized product detectable in pulse radiolysis experiment as a bump at 370nm at microseconds timescale. Have the authors considered such a possibility in the current study? I sincerely encourage authors to present some data/simulations or at least add a comment to this effect.

·       Can authors please include computational data for both stereoisomers for structures in Table 1? Also, it is important to have at least a CH3 group to mimic the sugar portion in the computational study for structures in Table 1. I strongly encourage you to report computational results with the CH3 or any suitable group to mimic sugar moiety.  The same applies to computational results reported in supplemental information.

·        Will the outcome of the experiment change significantly if the reaction is carried out under mildly acidic or basic conditions? Can authors comment on the pH sensitivity of the GC radical cation pair dynamics (Figure 8 – Keq species)?

·       Have the authors analyzed the products formed at the end of the radiolysis study to support their mechanistic hypothesis? What is the expected product after the formation of [G(-H+):C]- ?

Round 2

Reviewer 3 Report

The revised manuscript from Masi et. al. addresses all the points I had recommended in my previous review. Response letter from authors also highlights these changes.

I support the publication of this manuscript in Biomolecules without further revision.